# Maternal Lineages of Gepids from Transylvania

**DOI:** 10.3390/genes13040563

**Published:** 2022-03-23

**Authors:** Alexandra Gînguță, Bence Kovács, Balázs Tihanyi, Kitti Maár, Oszkár Schütz, Zoltán Maróti, Gergely I. B. Varga, Attila P. Kiss, Ioan Stanciu, Tibor Török, Endre Neparáczki

**Affiliations:** 1Department of Molecular Biology and Biotechnology, Faculty of Biology and Geology, Babeș-Bolyai University, 400006 Cluj-Napoca, Romania; gingutaalexandra@gmail.com; 2Department of Genetics, University of Szeged, H-6726 Szeged, Hungary; kovacs.bence.gabor@mki.gov.hu (B.K.); kitti.maar@gmail.com (K.M.); schutzoszi@gmail.com (O.S.); endre.neparaczki@bio.u-szeged.hu (E.N.); 3Department of Archaeogenetics, Institute of Hungarian Research, H-1014 Budapest, Hungary; tihanyi.balazs@mki.gov.hu (B.T.); zmaroti@gmail.com (Z.M.); varga.gergely@mki.gov.hu (G.I.B.V.); 4Department of Biological Anthropology, University of Szeged, H-6726 Szeged, Hungary; 5Department of Pediatrics and Pediatric Health Center, University of Szeged, H-6725 Szeged, Hungary; 6Faculty of Humanities and Social Sciences, Institute of Archaeology, Pázmány Péter Catholic University, H-1088 Budapest, Hungary; lordkisss@gmail.com; 7Institute of Archaeology and Art History, Romanian Academy, Cluj-Napoca Branch, 400084 Cluj-Napoca, Romania; istanciu2001@yahoo.fr

**Keywords:** mitogenome, Gepids, ancient DNA, NGS, migration period, population genetics

## Abstract

According to the written historical sources, the Gepids were a Germanic tribe that settled in the Carpathian Basin during the Migration Period. They were allies of the Huns, and an independent Gepid Kingdom arose after the collapse of the Hun Empire. In this period, the Carpathian Basin was characterized by so-called row-grave cemeteries. Due to the scarcity of historical and archaeological data, we have a poor knowledge of the origin and composition of these barbarian populations, and this is still a subject of debate. To better understand the genetic legacy of migration period societies, we obtained 46 full mitogenome sequences from three Gepid cemeteries located in Transylvania, Romania. The studied samples represent the Classical Gepidic period and illustrate the genetic make-up of this group from the late 5th and early 6th centuries AD, which is characterized by cultural markers associated with the Gepid culture in Transylvania. The genetic structure of the Gepid people is explored for the first time, providing new insights into the genetic makeup of this archaic group. The retrieved genetic data showed mainly the presence of Northwestern European mitochondrial ancient lineages in the Gepid group and all population genetic analyses reiterated the same genetic structure, showing that early ancient mitogenomes from Europe were the major contributors to the Gepid maternal genetic pool.

## 1. Introduction

The history of the so-called barbarian populations of the Migration Period is a subject of increasing interest, however, the reconstruction of the population history of these groups has been limited by the low number of reliable written and archaeological sources. From the migration period groups, Gepids had a great impact on the history of Central Europe, yet the investigation of this population so far has been marginal as their legacy is not claimed by any modern nations in Europe [1].

Gepids were one of the East Germanic groups which settled in the Carpathian Basin. Written sources first mention them in the second half of the 3rd century CE, and, based on archaeological data, they originated from the Wielbark Culture, from territory that is today Poland [1,2,3]; however, their presence in the Carpathian Basin can be evidenced only from the 5th century CE, related to the expansion of the Hun Empire they became a part of [1,2,3,4,5]. After the death of Attila (453 CE) and the collapse of the Hun power center in the Carpathian Basin, the Gepids rose and established a kingdom by acquiring territories on the Great Hungarian Plain, Sirmium, and Transylvania [1,2,3]. In the 6th century CE, their tense relationship with the Byzantine Empire and the Langobards of Transdanubia resulted in many battles. Finally, they were defeated by a Langobard–Avar alliance in 567 CE, and, consequently, the newly established Avar Khaganate occupied the former territory of the Gepid Kingdom [1]. From the archaeological point of view, the 5th-century-CE horizon of the Carpathian Basin is characterized by small grave groups and solitary burials with heterogeneous cultural backgrounds. The low number of graves was explained by a populational decrease connected to migration events and the collapse of the power center. Later, a demographic increase was assumed as the number of the 6th-century-CE findings is significantly higher. In the Gepid period, so-called row-grave cemeteries appeared showing analogues with the Merovingian Age burial grounds of Western Europe. Certain elements (artefact types and burial customs) of these burials were also detected in early Avar period cemeteries suggesting local populational continuity between the Gepid and Avar periods [1,3]. 

In the Transylvanian Basin, the same archaeological tendencies were registered, and two main types of row-grave cemeteries were distinguished [6]. The first, is usually referred to as the Morești group, represented by the early phase cemeteries dating to the 6th century CE. These series contain classic grave goods associated with the Gepid period population (e.g., bow-brooches with five knobs and chip-carved decoration, eagle-head buckles, spatha, long seax, and pear-shaped ceramic vessels) [6]. The second category, known as the Band–Vereșmort group is composed of mainly late row-grave cemeteries dating to the last third of the 6th century CE and first half of the 7th century CE. The use of these cemeteries extended into the early Avar period which is reflected in the characteristics of the archaeological material (e.g., lower number of bow-brooches, higher proportions of weapons, belt-sets, and pottery; appearance of horse burials). In addition, regional differences were registered concerning the archaeological material of the Gepid period cemeteries [6]. Unfortunately, many of these cemeteries are only partially excavated and published, therefore the sample selection and evaluation of the data is limited.

The European migration period is still underrepresented from the genetic point of view, restricted to a few data from populations associated with Avars [7,8,9,10,11], Lombards [12,13,14,15,16], Goths [17] and Huns [10]. So far, there is no genetic data available from the Gepid period. 

In this study we reveal the first 46 mitogenomes from three Gepid cemeteries located in Transylvania, Romania (Figure 1). All cemeteries are composed of burials that contained grave goods connected to the classic Gepid period population. Our aim was to investigate the genetic make-up of the Gepid period population in order to reveal their origin, population structure, and relationship with other ancient and modern Eurasian populations. To do so, we compared the full mtDNA sequences obtained in this study to all available ancient sequences from the public databases. Phylogenetic and population genetic analysis of the Gepid maternal lineages indicated Northwestern European affinity.

## 2. Materials and Methods

### 2.1. Archaeological Background

The late 5th century in the Transylvanian Basin was characterized by the appearance of row-grave cemeteries, or Reihengräberfelder, which were strongly connected to the Gepidic expansion [6]. In Transylvania, the archaeological research of the Gepidic period settlements began in the 1950s and since then, more than 100 sites were identified [18], but hitherto there is no genetic data available from any of these cemeteries. 

Migration Period cemeteries were often systematically robbed in attempt to recover their rich inventory. Gepid burials were frequently victims of near immediate robbing and disturbing of the deceased, which narrowed our sampling procedure considerably. Another problem regarding the Gepid cemeteries is that there are few cemeteries fully excavated and very limited information is published. In an attempt to carry out a representative sampling from the Classical Gepidic period, we chose 3 cemeteries which were excavated in the 21st century, using modern techniques. One of the cemeteries (Vlaha) is fully excavated and the other two are partially investigated. We collected teeth or petrous bones for DNA extraction from 50 individuals dated to the Classical Gepidic period from three row-grave cemeteries from Transylvania. We selected 7 individuals from the Carei–Bobald cemetery neighboring Northwestern Transylvania, 25 samples from the Șardu cemetery and 18 samples from Vlaha/Magyarfenes-Pad cemetery of which 14 mitogenomes were determined, both located in the North-Western Transylvanian Basin (Table 1). The selected graves include both females and males and burials with rich and poor grave goods. The archaeological description of the cemeteries and details of the archaeological and anthropological context together with the summary of phylogeographic affinity of the samples are presented in Appendix A and Appendix B. More details about the anthropological analyses and demographic structures of the Vlaha/Magyarfenes-Pad and Carei–Bobald cemeteries can be found in [19].

### 2.2. Sample Preparation and DNA Extraction

All methods preceding NGS were performed in the sterile laboratories dedicated for aDNA work at the Department of Archaeogenetics of the Institute of Hungarian Research and Department of Genetics, University of Szeged, Hungary.

Sampling was carried out using gloves and facemasks to minimize the risk of contamination with modern human DNA. From each individual, multi-root teeth or petrous bone was collected for the molecular analyses (Appendix A). Teeth were decontaminated with bleach and DNA was extracted from the teeth using the soaking method, which preserves the morphological integrity of teeth [20]. A pre-digestion step was performed in 3 mL extraction buffer containing 0.5 M EDTA and 100 µg/mL Proteinase K. Samples were incubated with the pre-digestion buffer for 30 min at 48 °C, followed by a 72-h digestion in extraction buffer containing 0.45 EDTA, 250 µg/mL Proteinase K, and 1% Triton X-100 at 48 °C. Then 12 mL of binding buffer containing 5 M GuHCl, 90 mM NaOAc, 40% isopropanol, 0.05% Tween-20 was added to the extract and DNA was purified on Qiagen MinElute columns. Quantity of DNA extracts were measured with Qubit 3.0 Fluorometer (Invitrogen) using the dsDNA High Sensitivity Assay kit.

### 2.3. NGS Library Preparation, Sequencing, and Hg Assignment

Library preparation, mitochondrial DNA capture, sequencing, and sequence analysis was done as described in [21]. Libraries were constructed from partial uracil-DNA glycosylase (UDG)-treated DNA extracts [22] using the double stranded library protocol [23] with double indexing [24] with the following modifications: the preamplification step was omitted and after adapter fill-in the libraries were directly double indexed in one PCR-step using Accuprime Pfx Supermix, containing 10 mg/mL BSA and 200 nM indexing P5 and P7 primers, in the following cycles: 95 °C 5 min, 12 times, 95 °C 15 s, 60 °C 30 s and 68 °C 3 s, followed by 5 min extension at 68 °C. 

All libraries were purified using MinElute columns and eluted in 20 µL EB buffer (Qiagen). DNA concentration was measured with Qubit 3.0 Fluorometer using the dsDNA High Sensitivity Assay kit and fragment size was determined using the TapeStation automated electrophoresis system (Agilent).

The endogenous human DNA content of each library was estimated by low coverage shotgun sequencing (Appendix A). Libraries with similar human DNA concentrations were combined, then mitogenomes were enriched as described in [25] and sequenced. 

The Cutadapt software [DOI:10.14806/ej.17.1.200] was used to trim the adapters from paired-end reads [26] and the quality of the reads was assessed with FastQC [27]. We conducted the further analysis only with the sequences that were longer than 25 nucleotides. The reads were mapped to the GRCh37.75 human genome reference sequence which contains the mtDNA revised Cambridge Reference Sequence (rCRS, NC_012920.1) using the Burrows Wheeler Aligner (BWA) v.0.7.9 software [28] with the BWA mem algorithm in paired mode and default parameters. The exogenous DNA was removed by applying an additional filter and we considered only molecules above 90% identity threshold to the reference genome. The binary alignment map (BAM) files were sorted and indexed using the Samtools v1.1 [29] and the PCR duplicates were removed using Picard Tools v1.113 [30]. Ancient DNA damage patterns were assessed using mapDamage 2.0 [31] with the read quality scores modified to account for post-mortem damage and sequences were visualized in Integrative Genomics Viewer v1.10.0 to identify potential errors during the SNP calling. Contamination was estimated using the Schmutzi algorithm [32] (Appendix A). Sample Vlaha1244 was removed from the analysis because of its high contamination, but the Vlaha1243 sample with 17% contamination was not removed, as its FASTA sequence from the “contaminating” molecules was assigned to the same Hg, indicating that all transitions were removed from these aDNA molecules due to UDG overtreatment. Mitochondrial haplogroups (Hg) were determined using HaploGrep2 (v2.4.0) [33] (Appendix A). The biological sex of the individuals was determined by two methods that uses shotgun sequencing reads, the X/Y ratio [34] and the endogenous DNA assigned to autosomes [35] (Appendix A). Additionally, we took extra care to exclude low MAPQ and pseudo autosomal regions of X and Y chromosomes. All details of mitogenome NGS data are reported in Appendix A. In just a few cases, the two methods did not give the same results.

The raw nucleotide sequence data of the 46 Gepid mitogenomes were deposited to the European Nucleotide Archive (http://www.ebi.ac.uk/ena, accessed on 21 February 2022) under the accession number: PRJEB50517.

### 2.4. Phylogenetic Study and Population Genetic Analyses

For the population genetic analysis, we merged all Gepids samples into one population. In order to determine the genetic distances between ancient populations mtDNA distribution of the studied samples were compared to populations from an archaic database containing 4324 ancient Eurasian mitogenomes [36], from which we used 92 populations based on time range and archaeological information (Appendix A). The database was augmented with the sequences obtained in this study. 

For building the median joining (MJ) networks [37], the selected sequences were aligned using MAFFT version 7 [38,39] with progressive G-INS-1 setting and converted to Nexus format using MEGA [40]. Phylogeographic connections were inferred from the geographic origin of the closest matching samples from the database (Appendix A, summarized in Appendix A). The phylogenetic analysis was preferably done with ancient DNA data, except for haplogroups for which ancient data were not available, where we used modern data. 

We used three independent methods for measuring the genetic similarity of Gepids to other ancient populations. In the first approach, we reduced the Hgs number to major Hgs, which decreased our population dataset to 45 dimensions, which is enough for showing the main correlations. The major Hgs frequencies were calculated and the principal component analysis (PCA) was performed using the “prcomp” function in R 3.6.3 [41].

In the second approach, we applied the traditional sequence-based method calculating pair-wise population differentiation values (Fst) with Arlequin 3.5.2.2 [42] from the whole mtDNA sequences, as described in [25]. For the Fst values calculation, a previously published database was used [36]. Multidimensional scaling (MDS) was applied on the matrix of linearized Slatkin Fst values [43] and visualized in a two-dimensional space using the cmdscale function implemented in R 4.1.2 [41].

Our third approach was measuring the so called shared haplogroup distance (SHD) values, as the presence of identical terminal sub-Hgs between populations testifies shared ancestry or past admixture events [44]. Pair-wise SHD distances were calculated between all the 93 ancient populations using the frequencies of the sub-Hgs, as described in [25]. 

## 3. Results

### 3.1. Sequencing Results and Haplogroup Assignment

Here we report 46 new mitogenome sequences (Appendix A) which were retrieved using the NGS method combined with mitogenome enrichment. Mitogenome DNA could not be obtained from 4 samples. The mitochondrial coverage was 0.2–2208×, with average coverage 375× (Appendix A). We obtained 38 mitogenomes above 5× coverage and 8 mitogenomes with less than 5× coverage (Appendix A). The lowest coverage sample, Vlaha0450, had 10.7% of the mitogenome covered 5× but Haplogrep could unambiguously assign the Hg of this sample, with all Hg defining SNP-s identified (Appendix A). Contamination was negligible (0–2%) in most of the samples, significant 17% contamination was indicated just in one sample. The 46 samples belong to 36 Hgs and 37 different haplotypes (Appendix A). We found a few identical haplotypes within cemeteries, which may indicate a potential direct maternal relationship of these individuals (Appendix A). 

#### 3.1.1. Haplogroup Composition and Phylogenetic Analysis

Gepid sequences cover almost the entire range of Western Eurasian Major-Hgs with the following frequencies: H 38%, HV 5%, I 8%, J 4%, N 3%, T 20% (T1 5% and T2 15%), U 11% (U3 3%, U 4 5% and U5 3%), and V 5% (Figure 2).

Phylogenetic relations were illustrated for each Hg using MJ Networks (Appendix A). The phylogenetic trees confirmed that the Transylvanian Gepid population belonged to western Eurasian lineages, most of them characteristic for Northwestern European regions. For example the following Hg-s cluster to the same node: The H1a Hg of Șardu03 and Șardu05 and a 7–9th century AD Anglo-Saxon individual (Network 3); the J1c2c1 Hg of Șardu26 and two 16th century AD samples excavated in Finland (Network 17); the T2f1a1 Hg of Sardu29 and several modern individuals from Denmark (Network 25); the T2b Hg of Carei111and a Langobard from Italy (Network 24); and the H1ba Hg of Carei35per2 and an Early Bronze Age sample from Germany (Network 4), indicating close maternal relation of at the Gepid samples to these Northwestern European individuals. 

A few lineages indicated Near Eastern affinity, such as individual Șardu18 (H13a2b), Carei35per1, (N1a1a1a*3) Vlaha0574, and Vlaha1216 (T1) and Vlaha0637 (T1a), which had the closest parallels from Bronze Age Anatolia and Late Chalcolithic Turkey. This Near Eastern affinity is most likely explained as the legacy of early European farmers [45,46]. 

The only lineage indicating east Eurasian affinity is the H6a1b Hg of the Carei34 individual (Appendix A, Network 7), which probably arrived with eastern immigrants of the Migration Period. 

Phylogenetic trees also revealed that many of the lineages had been present in the Carpathian Basin and the surrounding region from the Neolithic to the Middle Ages, which may indicate a potential population continuity of the 6th century population with earlier medieval groups in Transylvania, but a larger dataset is needed to prove this assumption.

#### 3.1.2. Kinship in the Gepid Cemeteries

We detected several identical haplotypes within all three cemeteries, which suggest possible direct maternal kinship between distant or neighboring burials. 

In the Vlaha/Magyarfenes–Pad cemetery we found 3 pairs of individuals with the same haplotype. All individuals came from an isolated group of burials in the northern part of the cemetery, which suggest a horizon of family burials in that area. We found rare subclades in these individuals, Vlaha1285 and Vlaha1293 belonging to H1ay*2 (Appendix A, Network 3), Vlaha1276 and Vlaha1288 belonging to H5a1g1a (Appendix A, Network 6), and Vlaha0547 and Vlaha1276 belonging to the T1 haplogroup (Appendix A, Network 22). The low frequency of these haplogroups makes it more likely that these individuals could be related. 

In the Șardu cemetery, we found a group of four individuals (Sardu08, 15, 76, and 77) with the same haplotype, belonging to the T2b5 subclade (Appendix A, Network 24), two individuals (Sardu03 and 05) belonging to the H1a subclade (Appendix A, Network 3) and other two individuals (Sardu80 and 89) belonging to the HV9 + 152 subclade (Appendix A, Network 13). The graves are positioned close to each other, suggesting that the burial place was used by a small community of related individuals. 

In Carei–Bobald cemetery, from the 7 investigated individuals we found two identical haplotypes (Carei40 and 41), from neighboring graves, belonging to the T2b4 subclade (Appendix A, Network 24), which were also very likely related.

#### 3.1.3. Population Genetic Analysis

We merged all 5–6th century Gepid lineages into a single population and measured their genetic distances from all ancient populations available in the public databases. We compared our data to published mitochondrial data [36] from 92 ancient Eurasian populations (Appendix A). PCA obtained from the major Hg frequencies places the Gepid population among Europeans (Figure 3). On Figure 3 our ancient samples map closest to Neolithic Europeans (EU_N), Central European Copper Age (HU_CA) and Bell Baker populations (BellBHu_CA), Poland Iron Age (Pol_IA) and Iceland Medieval (Ice_M).

To further confirm the genetic relationships of Gepids, an MDS plot was drawn from the linearized Slatkin Fst Values of the same populations, but for better resolution of the European side, 14 Asian populations were left out (Figure 4, Appendix A). The MDS plot also shows a clear grouping of Gepids with European populations, as they are closely clustered together in the left part of the figure with European Neolithic, Copper Age, and Iron Age groups, similar to the PCA plot results.

The SHD approach gave similar results, as the Gepids shared most identical Hg-s with ancient samples from Neolithic–Iron Age Europe (Appendix A). 

Fst and SHD analysis gave comparable results (Appendix A), as both indicated closest links to Anatolian Bronze Age (An_BA), the Balkan Neolithic (Balk_N) and Copper–Bronze Age (Balk_CBA) populations, Mediterranean Copper–Bronze Age (Mdt_CBA), Bell Baker groups from Germany (BellBGB_CA) and Great Britain (BellBG_CA), Near East Iran–Turan Copper Age (IrTur_CA), Near East Copper Age–Bronze Age (NE_CBA), Globular Amphora culture (GlobA_CA), Bronze Age Hungary (Hu_BA) and 10th century commoner population from Hungary (ConqC_M). Latter findings suggest that the overall mtDNA composition of the Gepid group from Transylvania may not differ significantly from other ancient groups of the Carpathian Basin.

## 4. Discussion

Using NGS technology combined with target enrichment, we described the maternal lineages associated with the Gepid culture in Transylvania. The genetic structure of a Gepid group is explored for the first time, providing new insights into the genetic structures of this archaic people. 

The retrieved genetic data showed mainly the presence of Northwestern European mitochondrial ancient lineages in the Gepid group, in line with the historically known ancient European geographic origin of these Germanic people. Regarding this, it is interesting to mention the affinities of the analyzed group with the population of the Wielbark culture described in [17], whose data were included in the present analysis under the name of the Poland Iron Age (Pol_IA) and the Langobards (Lang_M), Germanic groups with available mitogenome data. In Figure 3 and Figure 4 these groups map not very far from our Gepids, as their Fst and SHD values indicate close distances (Appendix A), though some of the values were not significant, probably due to sample size. These results are in agreement with the possible Wielbark origin of the Gepidic people, that they could have arrived to Transylvania from the northern region of East-Central Europe, today North and Central Poland [4].

The most frequent macrohaplogroup was H, followed by T, U, I, and HV. We detected just a single Asian lineage, indicating that the studied population did not significantly admix with immigrants from Asia. The Central-Northern European ancestry is dominant in the Gepid group and it is closest to Bronze Age populations from nearby regions. Due to the lack of genetic data from other periods in Transylvania we could not assess the relation with other local ancient individuals, but their affinity to the local population of the Carpathian Basin seems significant.

Each population genetic analysis reiterated the same genetic structure, showing that early ancient mitogenomes from Europe were the major contributors to the Gepid maternal genetic pool. Population genetic results also imply that the Gepid maternal ancestry may have derived mostly from local residents. This seems to contradict historical data, which describe the Gepids as Germanic immigrants from Northern Europe, however, these data do not exclude the possibility that many of the individuals could be immigrants from Northern Europe, as a comparable European maternal gene pool was present in both regions. 

The resolution provided by the mitochondrial genomes presented in this study cannot provide a full perspective of the demographic events. In order to depict with better resolution the migration and admixture events, we need to study these groups at the whole genome level, preferably with other contemporary samples from Transylvania and the surrounding regions. 

## Figures and Tables

**Figure 1 genes-13-00563-f001:**
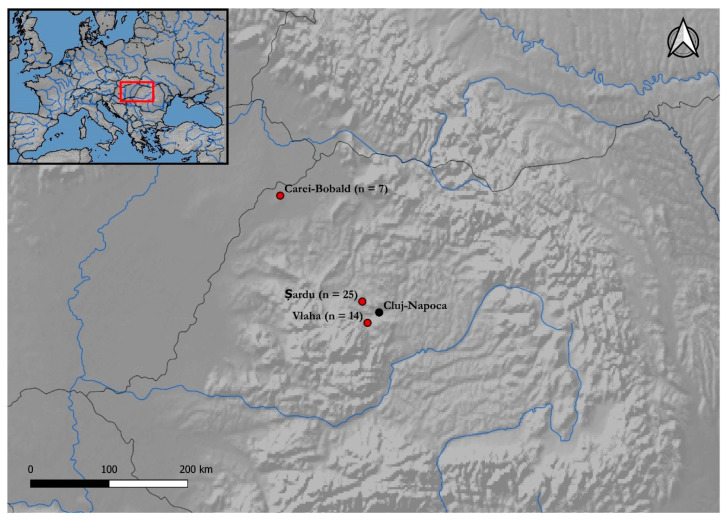
Location of the Gepid cemeteries under study. Sample size is indicated next to cemetery names. The map was generated using QGIS (references: QGIS Development Team QGIS Geographic Information System. Open Source Geospatial Foundation Project 2020.).

**Figure 2 genes-13-00563-f002:**
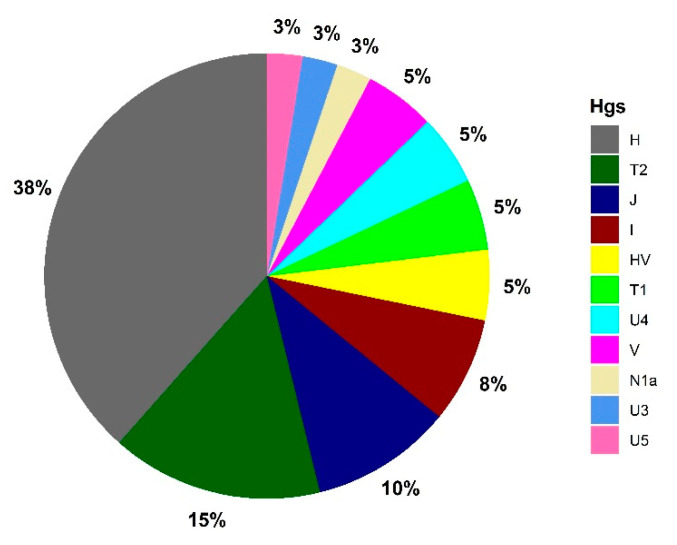
Major haplogroup distribution in the Gepid population from Transylvania.

**Figure 3 genes-13-00563-f003:**
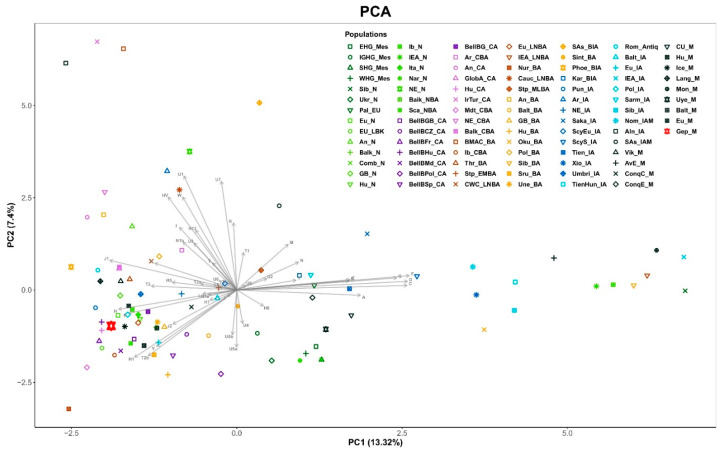
The principal component analysis (PCA) drawn from the major mtDNA haplogroup distributions of 93 Eurasian populations. The abbreviations of the population names are given in Appendix A.

**Figure 4 genes-13-00563-f004:**
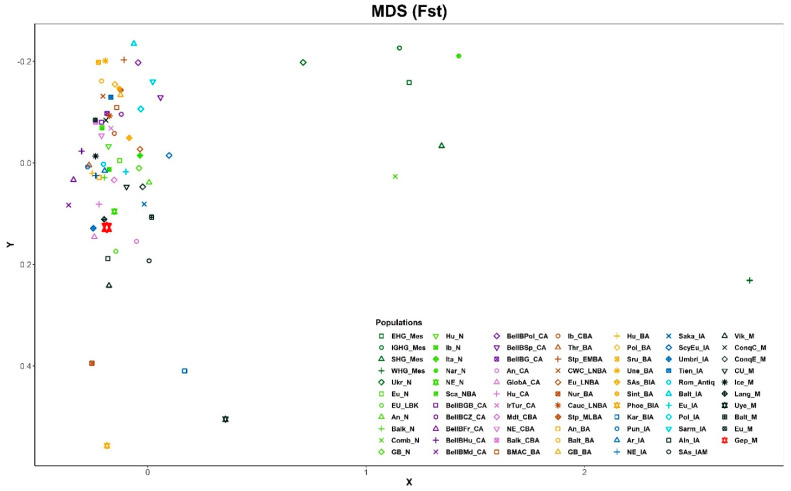
A multi-dimensional scaling (MDS) plot from the linearized Slatkin Fst values of Appendix A. For better resolution 14 Asian populations were left out from the analysis. The abbreviations of the population names are given in Appendix A.

**Table 1 genes-13-00563-t001:** Summary of the studied sample size from each cemetery.

Archaeological Site	Century AD	No. of Graves	No. of Collected Samples	No. of Mitogenomes Obtained
Carei–Bobald	6th	24	7	7
Șardu	5th–6th	49	25	25
Vlaha/Magyarfenes-Pad	6th	308	18	14

## Data Availability

The raw nucleotide sequences data were deposited to the European Nucleotide Archive (http://www.ebi.ac.uk/ena, accessed on 21 February 2022) under the accession number: PRJEB50517.

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
