# Peer review of "Maternal Lineages of Gepids from Transylvania"

_genes, 2022, doi:10.3390/genes13040563_

Round 1

Reviewer 1 Report

In the manuscript “Maternal Lineages of Gepids from Transylvania”, the authors obtain forty-six mitogenomes from three Gepid cemeteries located in Transylvania (Romania) and compare them to available datasets with the aim of characterizing the genetic make-up of the Gepids in order to reveal their origin, population structure and relationship with other ancient and modern Eurasian populations. In my opinion, lab procedures and bioinformatic analyses meet the standard for ancient DNA analysis, and the manuscript is well written. Potentially, this manuscript will be of interest to diverse fields, including paleogenomics, population genetics and archaeology. However, the authors should clarify a few aspects.

- The authors should state in the main text what was the minimum coverage for considering samples for analysis.

- Did the authors visually inspect the haplotypes? This could be helpful to identify potential error during the SNP calling, which can happen when qualities have been corrected with mapDamage to account for cytosine deamination.

- The authors state that “we reduced the Hgs number to major Hgs, which decreased our population dataset to 34 dimensions”. From Table S4, I can see that all individuals belonging to haplogroup H were collapsed together, as well, as other frequent haplogroups in Europe (J, K, T2 and U5). In my opinion, it would be more appropriate to consider a certain number of sub-haplogroups within each frequent haplogroup in order to get a clear picture of the European mtDNA diversity.

- Did the authors exclude from the analysis the sample with 17% contamination?

- The resolution of Figure 2 should be improved.

- In order to improve the availability of the data, it would be helpful to share the consensus mtDNA sequences in GenBank.

Reviewer 2 Report

The article « Maternal Lineages of Gepids from Transylvania » presents mitogenomes from a Middle Age population in Central Europe. The authors selected 50 samples from 3 Gepids cemeteries , extracted the DNA, enriched for mt DNA, and compared their haplogroups to published ancient data from the literature.
The article is sound, well written and pleasant. Archaeological characterization of the samples is precise. The protocoles followed, both in the lab or bioinformatically, are precisely described and followed up to date recommandations. 
I have a handful of questions :
1. Due to high migration rate of females, mitochondrial haplogroups are hardly discriminating between European populations. Indeed, we observe on figure 3 very low differentiation between post-Mesolithic European populations. Thus, the expectations of the authors are not very clear for me. In the introduction, it seems they want to differentiate 2 scenarios : local genetic continuity or population migration from an area that today is in Poland. But, as Iron Age polish populations belong to the same cluster as other European populations, it seems that the data here cannot discriminate between the two hypothesis.

2. The first axes of PCA (Fig 3) and MDS (Fig 4) are clearly drawn by the inclusion of Eastern Asian, Siberian populations, with high amount of A, B, C and D haplogroup, that may explain the lack of discrimination between European populations.

3. In the part 2.3, the authors describe their bioinformatic pipeline used for aligning the reads to the reference genome. They use a threshold of 25 bp, to prevent aligning environmental DNA to human genome. This threshold is a little low and may lead to spurious alignment. Usually, a 30 bp threshold is used

4. The authors determined the biological sex of the individuals by using the Ry statistics developed in Skoglund et al. This statistic compares the amount of reads aligning on the X and Y chromosome. However, it is not very precise, especially when the number of reads aligned on the human genome is low. The Rx statistic (Mittnik et al), that compares the coverage of the X and the autosomes, is much more reliable.

Round 2

Reviewer 1 Report

The authors have addressed all the suggestions. I think the paper should by published by Genes.